# Towards a Better Understanding of Beige Adipocyte Plasticity

**DOI:** 10.3390/cells8121552

**Published:** 2019-12-01

**Authors:** Esther Paulo, Biao Wang

**Affiliations:** Cardiovascular Research Institute, Department of Physiology, University of California, San Francisco, CA 94158, USA; Esther.PauloMirasol@ucsf.edu

**Keywords:** beige adipocyte, WAT browning, thermogenesis, metabolism

## Abstract

Beige adipocytes are defined as Ucp1^+^, multilocular adipocytes within white adipose tissue (WAT) that are capable of thermogenesis, the process of heat generation. In both mouse models and humans, the increase of beige adipocyte population, also called WAT browning, is associated with certain metabolic benefits, such as reduced obesity and increased insulin sensitivity. In this review, we summarize the current knowledge regarding WAT browning, with a special focus on the beige adipocyte plasticity, collectively referring to a bidirectional transition between thermogenic active and latent states in response to environmental changes. We further exploit the utility of a unique beige adipocyte ablation system to interrogate anti-obesity effect of beige adipocytes in vivo.

## 1. Introduction

Energy balance requires equivalent energy intake and energy expenditure. When energy intake exceeds energy expenditure, animals store excess energy as fat in adipose. When the adipose tissue is overloaded, excess fat is ectopically deposited in other metabolic tissues. The chronic energy excess can lead to obesity and further metabolic dysfunctions in other organs [1]. Adaptive thermogenesis, the heat production in response to changes of environmental temperature and diet, is a major contributor of total energy expenditure. Adaptative thermogenesis occurs mostly in brown fat [2], which contains specialized mitochondria-rich brown adipocytes whose thermogenic functionality is conferred by the uncoupling protein 1 (Ucp1) [3,4]. Two types of Ucp1^+^ adipocytes have been distinguished by their localization, developmental origin, and molecular signature in small rodents. The classical brown adipocytes are present in interscapular and perirenal adipose tissues, and they originate from Myf5^+^/Pax7^+^ skeletal muscle progenitors [5]. The beige adipocytes (also named brite adipocytes) are formed and clustered within subcutaneous white adipose tissue (WAT) in response to β-adrenergic stimulation [6,7,8]. The development and function of brown and beige adipocytes have been comprehensively reviewed elsewhere [9,10,11,12].

The presence of thermogenic, UCP1^+^ fat depots in adult humans has been recognized recently by ^18^F-fluoro-deoxy-glucose positron emission tomography (^18^F-FDG PET) with computer-assisted tomography (CT), due to their increased activity of glucose uptake [13,14,15,16]. ^18^F-FDG positive depots in humans contain both classical brown and beige adipocytes [17,18,19,20]. Several studies using anatomical and transcriptome profiling approaches have indicated that interscapular brown fat from human infants and neck brown fat from human adults share features with classical murine brown adipocytes, while supraclavicular fat of adult humans mainly consists of beige adipocytes. Importantly, the Ucp1^+^ beige adipocytes are steadily observed on humans or mice exposed to cold environment, but their appearances are also negatively correlated with aging and obesity, suggesting a potential role of beige adipocytes in regulating energy homeostasis.

## 2. Beige Adipocyte Formation: Different Players

It has been well established that activity of beige adipocytes is dominantly regulated by the sympathetic nervous system, induced by the cold or just the feeling of cold (Figure 1) [6,7,8]. Chemical denervation with 6-hydroxydopamina, prior to 3-week-old age or prior to cold exposure, impairs beige adipocyte formation, indicating the importance of sympathetic nerves during WAT browning [21]. Previous studies on anatomical positioning of sympathetic nerves in the subcutaneous inguinal white adipose tissue (iWAT), for example, by staining the tyrosine hydrolase^+^ (Th^+^) sympathetic nerves in fixed tissues [22,23], lack spatial and temporal precision [21,24,25]. Th^+^-sympathetic nerves and beige adipocytes are not distributed evenly across iWAT [22], suggesting that anatomic positioning of sympathetic nerves may also contribute to beige adipocyte formation. However, the actions of sympathetic nerves, the engagement of neurotransmitters, and their receptors at the contact sites of the sympathetic nerve and receiving cell in live mice are most unknown, and functional relevance of sympathetic nerve neurotransmission in beige adipocyte plasticity, especially during the postnatal period, is not fully elucidated yet.

Young et al. first observed the formation of brown-like adipocytes (multilocular in morphology and presence of the 32 kD mitochondrial uncoupling protein—Ucp1) in the parametrial fat pad, a traditional white fat pad, in female BALB/c mice after cold acclimation [26]. Numerous reports support the notion that these brown-like or “brite” adipocytes were transdifferentiated from existing white adipocytes [27,28,29,30,31]. In 2012, Wu from Spiegelman’s group established that these brown-like adipocytes (re-named as beige adipocytes) represented a distinctive population of thermogenic active adipocytes within WAT, in contrast to the classical brown adipocytes from the brown adipose tissue (BAT) [32]. And importantly, the clonal analysis of this study suggested distinct progenitors that can give rise to beige adipocytes, different with the transdifferentiation model. Follow-up studies did reveal that PDGFRα^+^ progenitors [33] and smooth-muscle lineage/mural progenitors [34,35,36] can differentiate into beige adipocytes in vitro and in vivo (Figure 1). MyoD^+^ progenitors also contribute to beige adipogenesis in the absence of βAR signaling [37], highlighting the heterogeneity of the beige adipocyte population. Notably, these two routes of beige adipocyte formation, de novo adipogenesis from progenitors and transdifferentiation from white adipocytes, are not mutually exclusive [25,36,38].

Other cell types within WAT also modulate cold-induced beige adipocyte formation via diverse mechanisms. The immune cell–beige adipocyte communications, involving macrophages (alternative activated macrophages and sympathetic neuron-associated macrophages), eosinophils, type 2 innate lymphoid cells (ILC2s), natural killer T cells (iNKTs), regulatory T cells (Tregs), and acetylcholine-producing CD45^+^ cells, may either positively or negatively regulate beige adipocyte formation, which have been reviewed elsewhere (Figure 1) [39]. These immune cells, along with gut microbiota remodeling, can also affect browning under changing nutrient status (such as intermediate fasting and dietary restriction [40,41,42]). Which route of WAT browning is triggered by immune cells remains to be characterized. In addition to cold stimulation, physiological process such as burn injury [43,44], exercise [45,46,47], cancer cachexia [48,49,50,51], lactation in females [52,53], and various circulating factors (hormones and metabolites) [45,54,55,56,57,58,59,60,61,62,63] also affect beige adipocyte formation, although the involvement of the sympathetic nervous system and the contribution of de novo beige adipogenesis or transdifferentiation (or both) in these settings have not be thoroughly determined. Beige adipocytes can be found in most, if not all, subcutaneous fat depots especially under cold adaptation [64]. For example, the subcutaneous inguinal white adipose tissue (iWAT) has been wildly studied in the field of beige adipocyte biology. Browning can occur in visceral fat depots under prolonged cold adaptation. In addition, multilocular Ucp1^+^ beige adipocytes can be found in other fat depots, such as perivascular adipose tissue and thigh adipose tissue [65].

## 3. Beige Adipocyte Plasticity: Ucp1^+^ vs. Ucp1^+^-Lineage

Beige adipocytes are defined as Ucp1^+^ multilocular adipocytes within WAT. Common practices in the aforementioned studies to detect WAT “browning” include quantitative determination of thermogenic gene expression (predominantly the *Ucp1* gene), immunoblot or immunostaining to detect Ucp1 protein, histological analysis of tissue sections to detect multilocular adipocytes, and Seahorse respiratory activity from tissues or cells or just isolated mitochondria. These methods do present a “screenshot” view of beige characteristics within WAT but fail to reveal the dynamics of beige adipocytes at cellular resolution and in chronological scale.

In 2013, Wolfrum’s laboratory elegantly demonstrated a bidirectional transition of beige adipocytes between thermogenic active and latent states in response to environmental changes in adult mice [66]. In the Ucp1-Tracer mice (*Ucp1-GFP:Ucp1-CreER:Rosa-tdRFP*), Ucp1 expression and Ucp1 lineage (regardless of current Ucp1 expression) were visualized by GFP and RFP separately. Under warm condition, RFP^+^:GFP^+^ beige adipocytes became latent RFP^+^:GFP^−^ white-like unilocular adipocytes, the process of beige adipocyte whitening. These unilocular RFP^+^ (being Ucp1^+^ in the past) adipocytes transformed into multilocular GFP^+^ beige adipocytes again upon cold stimulation, strongly suggesting that the Ucp1^+^ adipocytes can manifest either beige or white adipocyte phenotype, depending on ambient temperature.

We [25] and others [67,68] have evaluated beige adipocyte formation during postnatal development. *Ucp1* expression in iWAT was absent in 1-week-old pups, reached a peak in 3-week-old pups, and gradually reduced to base line in 8-week-old adult mice. *Ucp1* expression correlated with beige adipocyte abundance; H&E staining confirmed the presence of the multilocular beige adipocytes only in 3-week-old pups. These beige adipocytes formed during postnatal development were referred as postnatal beige adipocytes. Lineage tracing experiments in Ucp1-Cre:Rosa-STOP-mT/mG mice showed that GFP^+^ Ucp1^+^-lineage adipocytes were still present in adults, but being latent (unilocular in morphology and negative for Ucp1 expression) [25]. Thus, the Ucp1^+^ beige adipocytes formed postnatally (~3 weeks of age) in iWAT gradually lose their beige characteristics (Ucp1 expression and multilocular morphology) in adult stage, but they can reappear in response to cold [25,67,69,70].

We employed Ucp1-Cre inducible diphtheria toxin receptor (iDTR)-mediated cell ablation system (Figure 2a,b) [71], to determine whether postnatal beige adipocytes and cold-induced beige adipocytes at adult stage were the same cell population. In 3-week-old Ucp1-Cre:Rosa-STOP-iDTR (abbreviated as Cre^+^) pups, the postnatal beige adipocytes were successfully ablated three days after diphtheria toxin (DT) administration, confirmed by absence of *Ucp1* expression and multilocular adipocytes. Then, we injected CL316243 (a β3 specific agonist, abbreviated as CL) for 4 or 7 consecutive days in control (Rosa-STOP-iDTR, abbreviated as Cre^−^) and Cre^+^ mice at 8 weeks of age. Although CL strongly induced *Ucp1* mRNA levels in iWAT from Cre^−^ mice, it had no effect in beige-ablated Cre^+^ mice. A similar result was obtained in 4-day 8 °C cold stimulation. Longer cold stimulation (7- and 14-day 8 °C) induced *Ucp1* expression in beige-ablated Cre^+^ mice. This could be due to compensatory beige adipocyte adipogenesis [35,36,72]. In contrast, both CL and cold induced epididymal WAT (eWAT) *Ucp1* expression similarly in postnatal beige adipocyte-ablated and non-ablated mice [25], suggesting the involvement of de novo beige adipogenesis in cold-induced beige adipocyte formation in visceral epididymal fat depot.

Thus, two consecutive phases of beige adipocyte formation in iWAT are present during postnatal development; the first one is due to de novo adipogenesis (genesis) and peaks at 3 weeks of age, and the other one is through restoring the beige characteristics in the latent beige adipocytes under β-adrenergic stimulation at adult stage (renaissance, the restoration of thermogenically active beige adipocytes) (Figure 3). This “beige adipocyte renaissance” model emphasizes that white adipocyte be converted to beige adipocyte by cold has Ucp1 expression history [25], which is supported by observations from other laboratories [38,68,69,73].

To avoid confusion with preexisting literatures, here we describe adipocytes with Ucp1 expression history (regardless of their cellular morphology or Ucp1 expression currently) as Ucp1^+^-lineage adipocytes, and white adipocytes without Ucp1 expression history as Ucp1^−^-lineage adipocytes. Beige adipocyte plasticity collectively refers to the ability of Ucp1^+^-lineage adipocytes to transform between thermogenic latent (unilocular and Ucp1^−^) and active (multilocular and Ucp1^+^) states in accordance to environmental fluctuations (Figure 3), including the maintenance of active beige adipocytes from postnatal (3 weeks of age) to adult stage (≥8 weeks of age), and cold-induced beige adipocyte renaissance and their maintenance in adults under different nutritional status. Notably, only latent Ucp1^+^-lineage adipocytes, not Ucp1^−^-lineage white adipocytes, can be transdifferentiated into active Ucp1^+^ beige adipocytes by cold in adult mice.

## 4. HDAC4 Signaling in Ucp1^−^-Lineage White Adipocytes Controls Ucp1^+^-Lineage Beige Adipocyte Plasticity Non-Cell Autonomously

One critical obstacle in beige adipocyte research is that there are no specific Cre lines to precisely target white, brown, or beige adipocyte (present or past) in mouse models. Tissue specificity of available Cre lines has been summarized [74,75,76]. Caution should be taken to link the WAT browning to the systemic metabolism in mouse models using *Adiponectin, Prx1* or *Fabp4,* or *Ucp1* enhancer/promoter-driven gene deletion or overexpression [24,25,77,78]. With this notion in mind, the maintenance of the thermogenically active Ucp1^+^-lineage adipocytes in iWAT has been investigated in several groups. For example, Kajimura’s group demonstrated that mitophagy is essential to maintain mitochondrial functionality in the active beige adipocytes, as Ucp1-Cre-driven deletion of Atg5 or Atg7 in Ucp1^+^-lineage adipocytes led to their accelerated disappearance [79]. Transcriptional factors or cofactors, such as PRDM16, LSD1, GR, and ZFP423, also regulate the maintenance of the Ucp1^+^-lineage adipocytes upon aging and changing temperature (Table 1) [38,73,80,81,82]. Because Adiponectin-Cre, rather than Ucp1-Cre, is used in these studies, it is unclear whether they regulate beige adipocyte maintenance cell autonomously (in Ucp1^+^-lineage beige adipocytes themselves) or non-cell autonomously (in Ucp1^−^-lineage white adipocytes).

We further investigated the cellular and molecular mechanisms that control beige adipocyte plasticity (Figure 4) [21,24,25]. We showed that blocking cAMP production in white adipocytes, but not in Ucp1^+^ beige adipocytes themselves, fully abolished cold-induced beige adipocyte formation in iWAT (Figure 4a) [21], highlighting the role of cAMP signaling in white adipocytes during cold-induced beige adipocyte formation. Previously, we identified that salt-inducible kinases (SIKs) can suppress cAMP-dependent transcription by phosphorylating and inactivating two types of transcription cofactors: CREB-regulated transcription coactivators (CRTCs) and class IIa HDACs [83,84]. Removing SIK activity specifically in pan adipocytes by deleting their upstream activating kinase, liver kinase b 1 (Lkb1), led to persistent beige adipocyte renaissance and maintenance independently of cold stimulation, which was not observed in brown adipocyte-specific Lkb1 knockout mice (Figure 4b) [25]. Consistently, the global Sik1:Sik2 double knockout mice showed elevated *Ucp1* expression in iWAT [25], although the precise roles of SIKs in this setting have not been thoroughly addressed in a tissue-specific manner. However, deleting HDAC4, one substrate of SIKs, in white adipocytes reversed the beige adipocyte renaissance phenotype in pan adipocyte-specific Lkb1 knockout (Lkb1^AKO^) mice [25]. On the other hand, expressing a constitutively active HDAC4 in adipocytes (H4-TG mice) phenocopied the Lkb1^AKO^ mice, regarding the beige adipocyte plasticity in iWAT (Figure 4a,b) [24]. Importantly, ablating the preexisting beige adipocytes at 3 weeks of age prevented cold-induced beige adipocyte formation in adult H4-TG mice [24].

Collectively, these studies in the aforementioned mouse models clearly demonstrate that cold-induced activation of class IIa HDACs in Ucp1^–^-lineage white adipocytes are necessary and sufficient to drive beige adipocyte plasticity non-cell autonomously, suggesting a to-be-characterized crosstalk between Ucp1^−^- and Ucp1^+^-lineage adipocytes as the key regulatory axis (Figure 4a,b).

## 5. Beige Adipocytes in Metabolic Diseases

Isolated mitochondria from beige adipocytes in iWAT and brown adipocytes in interscapular brown adipose tissue (iBAT) process similar thermogenic capacities in vitro [4]. Based on the calculation of Ucp1 protein or mRNA abundance, it is estimated that maximumly activated beige adipocytes process about a quarter of total Ucp1-dependent thermogenic activity in mice [85]. Thus, beige adipocytes, similar to brown adipocytes, can potentially promote energy expenditure, although direct evidence is still lacking [86,87].

Beige adipocyte plasticity is also modulated by nutrient status; excess energy intake (e.g., by high-fat diet (HFD)) induces beige adipocyte whitening [9]. Thus, the multilocular Ucp1^+^ beige adipocytes present under normal chow cannot retain their beige characteristics under HFD. Consequently, it will be unlikely that metabolic benefits are contributed by thermogenic activity from these bona fide beige adipocytes, because they are neither multilocular nor Ucp1-positive after long-term HFD. For example, very few mouse models show persistent WAT browning after long-term HFD, with the exception of the brown adipocyte-specific Tle3 and Tfam knockout mice and global Irf3 knockout mice (Table 1) [77,88,89]. The lack of stable and thermogenically active beige adipocytes under obesogenic conditions still represents a significant obstacle in advancing beige adipocyte-based therapies, which also reflects an incomplete understanding of the mechanisms governing beige adipocyte formation and function, despite over 30 years of research in this field. Thus, future studies are warranted to investigate additional regulatory mechanisms controlling beige adipocyte plasticity.

**Table 1 cells-08-01552-t001:** Phenotype summary in selective mouse models with WAT browning phenotype under normal chow and housed at room temperature, focusing on genetic manipulations via Fabp4, adiponectin, and Ucp1 promoters.

Moues Model	Genotype	Mechanism of Action		WAT Browning	Ref.
		Ucp1^+^-Lineage	Ucp1^−^-Lineage		
Gnas^BKO^	*Ucp1-Cre:Gnas^f/f^*	cAMP LOF	NC	Normal	[21]
Gnas^AKO^	*Adiponectin-Cre:Gnas^f/f^*	cAMP LOF	cAMP LOF	Absent	[21]
Lkb1^BKO^	*Ucp1-Cre:Lkb1^f/f^*	Lkb1 LOF	NC	Normal	[25]
Lkb1^AKO^	*Adiponectin-Cre:Lkb1^f/f^*	Lkb1 LOF	Lkb1 LOF	Enhanced	[25]
Hdac4^BKO^	*Ucp1-Cre:Hdac4^f/f^*	Hdac4 LOF	NC	Normal	[21]
Hdac4^AKO^	*Adiponectin-Cre:Hdac4^f/f^*	Hdac4 LOF	Hdac4 LOF	Normal	[25]
Lkb1;Hdac4^AKO^	*Adiponectin-Cre:Lkb1^f/f^:Hdac4^f/f^*	Hdac4 LOF	Hdac4 LOF	Normal	[25]
H4-TG	*Fabp4-HDAC4.3A*	HDAC4 GOF	HDAC4 GOF	Enhanced	[24]
H4-TG;Ucp1-iDTR	*Ucp1-Cre:Rosa-LSL-iDTR:Fabp4-HDAC4.3A*	Ablated	HDAC4 GOF	Absent	[24]
P16-TG	*Fabp4-Prdm16*	Prdm16 GOF	Prdm16 GOF	Enhanced	[8]
Prdm16^AKO^	*Adiponectin-Cre:Prdm16^f/f^*	Prdm16 LOF	Prdm16 LOF	Absent or Normal	[38,82]
Ebf2-TG	*Fabp4-Ebf2*	Ebf2 GOF	Ebf2 GOF	Enhanced	[90]
Zfp423^iAKO^	*Adiponectin-rtTA:TRE-Cre:Zfp423^f/f^*	Zfp423 LOF	Zfp423 LOF	Enhanced	[81]
Lsd1^AKO^	*Adiponectin-Cre:Lsd1^f/f^*	Lsd1 LOF	Lsd1 LOF	Absent	[80,91,92]
Lsd1-cTG	*Adiponectin-Cre:CAG-LSL-Lsd1*	Lsd1 GOF	Lsd1 GOF	Enhanced	[80,93]
Zfp516-TG	*Fabp4-Zfp516*	Zfp516 GOF	Zfp516 GOF	Enhanced	[91,94]
Zfp516-TG; Lsd1^AKO^	*Fabp4-Zfp516:Adiponectin-Cre:Lsd1^f/f^*	Zfp516 GOF + Lsd1 LOF	Zfp516 GOF + Lsd1 LOF	Absent	[91]
Hdac3^AKO^	*Adiponectin-Cre:Hdac3^f/f^*	Hdac3 LOF	Hdac3 LOF	Absent or Normal	[95,96]
Raptor^AKO^	*Adiponectin-Cre:Raptor^f/f^*	mTORC1 LOF	mTORC1 LOF	Absent or Normal or Enhanced	[97,98,99]
Rheb^AKO^	*Adiponectin-Cre:Rheb^f/f^*	cAMP GOF	cAMP GOF	Enhanced	[100]
ADM2-TG	*Fabp4-ADM2*	ADM2 GOF	ADM2 GOF	Enhanced	[101]
mitoNEET-TG	*Fabp4-mitoNEET*	mitoNEET GOF	mitoNEET GOF		[102]
Tle3^AKO^	*Adiponectin-Cre:Tle3^f/f^*	Tle3 LOF	Tle3 LOF	Enhanced	[88]
Tfam^BKO^	*Ucp1-Cre:Tfam^f/f^*	Tfam LOF	Tfam LOF	Enhanced	[77]
BMP8b-TG	*Fabp4-BMP8b*	BMP8b GOF	BMP8b GOF	Enhanced	[103]

BKO: Brown adipocyte-specific knockout mice mediated by Ucp1-Cre. AKO: Adipocyte-specific knockout mice mediated by Adiponectin-Cre. LOF: Loss-of-function. GOF: Gain-of-function.

Thermoneutrality (~30 °C for mice) is the ambient temperature that the animal maintains its body temperature through its basal metabolism without physical and thermogenic activities, while adaptive thermogenesis from brown and beige adipocytes is already (partially) activated at room temperature (RT; ~22 °C) [104]. Thus, comparison of metabolic phenotypes in mice housed at RT and thermoneutrality constitutes an experimental setup to determine the contribution of adaptive thermogenesis to total energy expenditure. For example, the adipocyte-specific Carnitine Palmityltransferase 2 (Cpt2; an obligate step in mitochondrial long-chain fatty acid oxidation) knockout mice exhibits defective fatty acid oxidation and adaptive thermogenesis at both RT and thermoneutrality [105,106]. These mice exhibit reduced adiposity only at RT, suggesting that decreased adaptive thermogenesis can contribute to the development of obesity in these mice at RT [105,106]. Similarly, adipocyte BMP8b transgenic mice have been shown to sustain beige adipocytes at both RT and thermoneutrality due to the remodeling of neuro-vascular network, only manifesting reduced susceptibility to HFD-induced obesity at RT [103]. 

However, Kozak’s group has recently shown that Ucp1^+^-lineage beige adipocytes appear postnatally (~3 weeks of age), regardless of housing temperature, and they remain present in iWAT (even in mice raised from birth at 30 °C) [70]; therefore, experiments at thermoneutrality do not fully exclude potential contribution of Ucp1^+^-lineage beige adipocytes to systemic metabolism. We targeted beige adipocyte plasticity in order to maintain functional beige adipocyte population under HFD condition. We showed that activation of HDAC4, by removing its upstream suppressor Lkb1 (in Lkb1^AKO^ mice) or overexpressing constitutively active HDAC4 (in H4-TG mice), in white adipocytes can maintain the presence of the multilocular Ucp1^+^ beige adipocytes in iWAT after a 4–5-week HFD [24,25]. However, these beige adipocytes are no longer seen after 10–12 weeks of HFD. These observations suggest two possibilities: (1) Improved metabolic performance in Lkb1^AKO^ and H4-TG mice after long-term HFD may be merely due to the HDAC4 activation in white adipocytes, regardless of beige adipocyte expansion; or (2) these Ucp1^+^-lineage adipocytes can modulate systemic metabolism, even in latent stage (being Ucp1^−^ and unilocular).

To circumvent this concern, we characterized the Ucp1-iDTR system that enables us to address functions of these Ucp1^+^-lineage beige adipocytes (regardless of their current status of Ucp1 expression) in mice at RT (Figure 2a,b) [25]. After diphtheria toxin (DT)-induced ablation of both Ucp1^+^-lineage brown and beige adipocytes, brown adipocytes in iBAT are fully regenerated in 3–5 weeks at RT, beige adipocytes in the iWAT remain absent, and tissue inflammation triggered by DT-induced cell ablation in both BAT and iWAT is fully resolved [24,25]. Using this Ucp1^+^-lineage beige adipocyte ablation system [25], we attempted to determine specific functions of beige adipocytes by genetic ablating in the H4-TG mice [24]. We found that beige adipocyte ablation did reverse the reduced adiposity but did not improve insulin sensitivity in H4-TG mice during initial weeks of HFD (up to 5 weeks). However, this effect was absent after 10 weeks of HFD, which is consistent with the observation that multilocular and Ucp1^+^ beige adipocytes were no longer present in H4-TG after 10 weeks of HFD [24]. Of note, recent studies suggest that Ucp1-independent thermogenic mechanisms, such as creatine-driven substrate cycle and Serca2b-mediated calcium cycle, are present, which may also affect systemic metabolism [107,108]. Whether Ucp1^+^-lineage adipocytes, at either active or latent state, process these compensatory thermogenic mechanisms remains undetermined.

## 6. Conclusions and Future Perspectives

The list of WAT browning-promoting agents is burgeoning [9,85,109], although whether they promote beige adipogenesis from progenitors or renaissance and maintenance of Ucp1^+^-lineage beige adipocytes, or both, is unknown. Our novel Ucp1^+^-lineage adipocyte ablation system can be applied here to determine the contribution of these Ucp1^+^-lineage adipocytes in the WAT browning process that is induced by these agents. As a proof-of-concept, we showed that the WAT browning effect of CL316,423 is absent in mice after the ablation of Ucp1^+^-lineage adipocytes [25]. Identifying the targeted cell types (progenitors, Ucp1^+^ multilocular beige adipocytes or Ucp1^–^ unilocular white adipocytes) of these browning-promoting agents will present the first step to evaluate their mechanism, efficacy, and potential side-effect for clinical applications. For example, both cold and Mirabegron (an orally active β3-adrenergic receptor agonist) can induce WAT browning but accelerate atherosclerotic plaque growth and instability in ApoE and Ldlr knockout mice [110,111]. This beige adipocyte ablation approach will be able to elucidate whether WAT browning indeed has detrimental effect in atherosclerosis.

The association between WAT browning and metabolic benefits does not necessarily prove the causality relationship of the two: Whether WAT browning indeed improves systemic metabolism is not known. In theory, this Ucp1^+^-lineage adipocyte ablation system will allow us to address this question by comparing metabolic phenotypes before and after beige adipocyte ablation in mouse models with persistent WAT browning. Importantly, this approach will not discriminate the thermogenic (Ucp1-dependent and independent) and non-thermogenic contributions of beige adipocytes to systemic metabolism. Notably, white adipocytes are much more abundant than active beige adipocytes and progenitors in WAT, implying that targeting white adipocytes to promote beige adipocyte maintenance indirectly may represent a more practical therapeutic means to prevent or even treat obesity and associated metabolic disorders.

## Figures and Tables

**Figure 1 cells-08-01552-f001:**
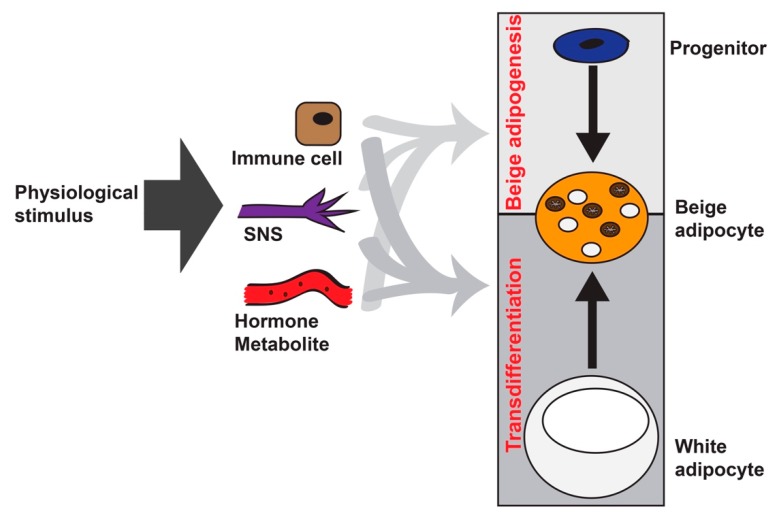
Players that are involved in beige adipocyte formation. There are two major routes to generate the Ucp1^+^ and multilocular beige adipocytes in white adipose tissue (WAT) by various physiological stimuli, such as cold, nutrient excess, burn injury, exercise, and cancer cachexia. The first route is through de novo beige adipogenesis from PDGFRα^+^, or PDGFRb^+^ or MyoD^+^ progenitors residing in WAT. The other route is through transdifferentiation from existing white adipocytes. Although the sympathetic nervous system (SNS) is the major driver for beige adipocyte formation, various types of immune cells and circulating hormones (thyroid hormone, Natriuretic peptides, FGF21, Irisin, IL-6) and metabolites (succinate, lactate, and bile acid) can influence beige adipocyte formation via either route.

**Figure 2 cells-08-01552-f002:**
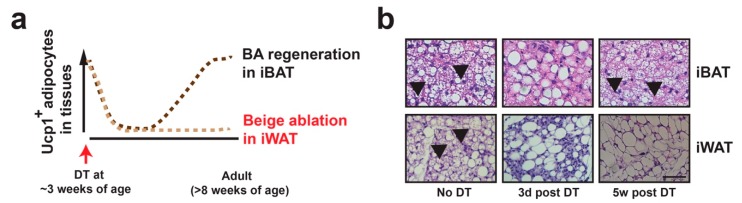
Beige adipocyte specific ablation in vivo. (**a**) Diagram showing beige adipocyte ablation strategy. In Ucp1-Cre:Rosa-STOP-iDTR mice at room temperature, DT injection at ~3 weeks of age induces cell death of Ucp1^+^ brown adipocytes in interscapular BAT (iBAT) and beige adipocytes in iWAT within 3 days. Brown adipocytes gradually repopulate the BAT in 3–5 weeks, while Ucp1^+^ beige adipocytes remain absent in iWAT. (**b**) Representative H&E staining of iBAT and iWAT from Ucp1-Cre:Rosa-STOP-iDTR^het^ mice prior to 3 days or 5 weeks post-DT injection. Scale bar: 200 μm. Black arrows: Multilocular brown and beige adipocytes.

**Figure 3 cells-08-01552-f003:**
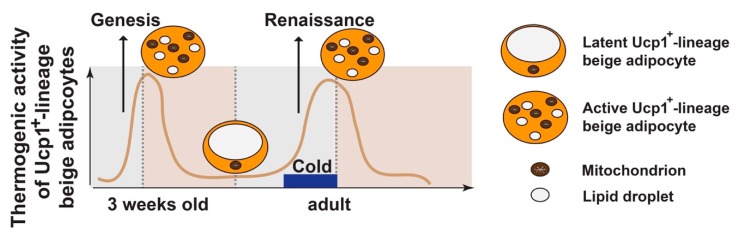
Beige adipocyte plasticity. During postnatal development, thermogenic active beige adipocytes (positive for Ucp1 and multilocular in morphology) in iWAT peak at ~3 weeks of age, due to de novo beige adipogenesis from progenitors (“genesis”). These Ucp1^+^-lineage beige adipocytes are unstable by default, becoming latent beige adipocytes (negative for Ucp1 and unilocular in morphology) at adult stage (~8 weeks of age) at room temperature (RT). Under 4 °C cold stimulation, their beige characteristics are restored (“renaissance”). The “maintenance” of beige adipocytes refers to the process that thermogenic active beige adipocytes remain their thermogenic activity, which requires persistent cold stimulation. Withdrawal from cold or nutrient excess can convert the active beige adipocytes back to the latent stage.

**Figure 4 cells-08-01552-f004:**
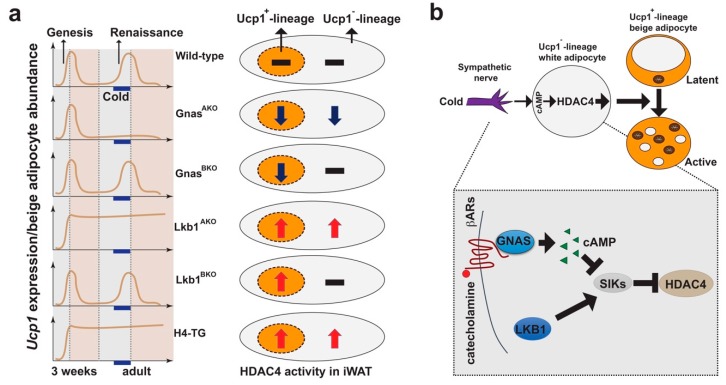
Central role of HDAC4 in white adipocytes during beige adipocyte plasticity. (**a**) Left: Phenotypes of beige adipocyte genesis and renaissance in mouse models. Right: Blackline, dark-blue down arrow and red up arrow indicate HDAC4 activity in Ucp1^+^ and Ucp1^−^ lineage adipocytes. (**b**) Proposed model: Beige adipocyte plasticity, including renaissance and maintenance, is mainly regulated by cAMP- and HDAC4-dependent signaling in Ucp1^−^-lineage white adipocytes in a non-cell autonomous fashion. Shaded box: Diagram showing salt-inducible kinases (SIKs)–HDAC4 pathway in white adipocytes. Upon cold-stimulation, catecholamine released by sympathetic nerves binds to the beta-adrenergic receptors (βARs) to induce cAMP production by activating adenylate cyclase-stimulating G alpha subunit (GNAS). cAMP inhibits SIKs, leading to dephosphorylation and nucleus-translocation of class IIa HDACs (such as HDAC4). Liver kinase b 1 (LKB1) phosphorylates and activates SIKs, causing phosphorylation and nuclear exclusion of class II HDACs. Abbreviations: Ucp1: Uncoupling protein 1; iWAT: Inguinal white adipose tissue; cAMP: Cyclic adenosine monophosphate; HDAC4: Class IIa histone deacetylase 4.

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
