# Peer review of "Towards a Better Understanding of Beige Adipocyte Plasticity"

_cells, 2019, doi:10.3390/cells8121552_

Round 1
Reviewer 1 Report
This review prepared by the Biao Wang laboratory summarizes the current understand of beige adipocyte plasticity. The review is well-balanced and updated, such that the paper will be well received by peers in the field. A few suggestions will improve the content and message of the review.
1) The authors should address how pathological circumstances in humans and mice impact beige fat cell plasticity. For example, how does cell plasticity contribute to burn responses (Cell Metab. 2016 23(6):1200-1206), hyperlipidemia (Proc Natl Acad Sci USA 2019 116(22):10937-10942; Cell Metab. 2013 18(1):118-290), and cachexia (Nat Med. 2016 22(10):1120-1130; Cell Metab. 2014 20(3):433-47; Cell Metab. 2016 23(2):315-23; Nature. 2014 513(7516):100-4).
2) Macrophage-derived TH does not contribute to thermogenesis (Nat Med. 2017 23(5):623-630). The authors might add some insight into how immune signals contribute to beige adipocyte plasticity.
3) Lastly, the term 'renaissance' should be more clearly defined in the text. It's not a commonly-used scientific term.
Author Response
1) The authors should address how pathological circumstances in humans and mice impact beige fat cell plasticity. For example, how does cell plasticity contribute to burn responses (Cell Metab. 2016 23(6):1200-1206), hyperlipidemia (Proc Natl Acad Sci USA 2019 116(22):10937-10942; Cell Metab. 2013 18(1):118-290), and cachexia (Nat Med. 2016 22(10):1120-1130; Cell Metab. 2014 20(3):433-47; Cell Metab. 2016 23(2):315-23; Nature. 2014 513(7516):100-4).
These studies have been included in revised manuscript. How these processes affect WAT browning, thorough either de novo beige adipogenesis or transdifferentiation from white adipocytes or both, requires further investigations.2) Macrophage-derived TH does not contribute to thermogenesis (Nat Med. 2017 23(5):623-630). The authors might add some insight into how immune signals contribute to beige adipocyte plasticity.
It is possible that both routes, de novo beige adipogenesis and transdifferentiation from white adipocytes, may be involved.Notably, there are many outstanding reviews that comprehensively cover one or many aspects of the beige adipocyte biology. Our manuscript is specifically focused on the beige adipocyte plasticity. Therefore, the immune cell-beige adipocyte communication is not thoroughly discussed here. The issue regarding the roles of macrophages in WAT browning is in-depth summarized in the recent review in Cell Metabolism 2018 (one year after the Nature Medicine publication).3) Lastly, the term 'renaissance' should be more clearly defined in the text. It's not a commonly-used scientific term.
We define beige adipocyte renaissance as the restoration of thermogenically active beige adipocytes.Reviewer 2 Report
First of all, I think the review work presented by Paulo E, and Wang B, is a good summary of studies concerning beige adipocytes formation and beige adipocyte plasticity.
This review is very interesting and will contibute to understand of beige adipocyte plasticity. It represents an important compendium of information about “beiging”, and I think it is necessary in this scientific area. In particular, the authors detail many tools in basic research to identify different UCP1+ cell populations, and the posible utility to search physiologycal roles in beige adipocytes in thermogenesis and metabolism. Thus, I recommend this manuscript as a suitable review for publication in the “Cells” with a minor comments.
In abstract section in line 10: The phenomenon of the “expansión” should be changed for” increase”
In line 48-50: authors summarize the importance of sympahtetic nerves (TH positive), in contribution of beige adipocyte formation. Around 3-week of age, How are the innervations? its known if Th cells are missing or less does the amount of UCP1 change at 3-week age?
I think the authors should add a sentence about this relationship.
In line 58: “Young first”….should be changed for ” Young et al..in … “ first observed
In lines 78-81, the relationship the nutrient status and browning was summarized. Its possible that the quality of breast-feeding after 3 week of birth, could be related to an increase of UCP1+ levels during this period?
UCP1+ is misspelled in figure 3.
Author Response
In abstract section in line 10: The phenomenon of the “expansión” should be changed for” increase”
In line 48-50: authors summarize the importance of sympahtetic nerves (TH positive), in contribution of beige adipocyte formation. Around 3-week of age, How are the innervations? its known if Th cells are missing or less does the amount of UCP1 change at 3-week age?
I think the authors should add a sentence about this relationship.
In line 58: “Young first”….should be changed for ” Young et al..in … “ first observed
In lines 78-81, the relationship the nutrient status and browning was summarized. Its possible that the quality of breast-feeding after 3 week of birth, could be related to an increase of UCP1+ levels during this period?
UCP1+ is misspelled in figure 3.
These minor concerns have been addressed in the revised manuscript. Suggestions about correlation between Ucp1 expression at 3-week of age and sympathetic nerve density or change of feeding patter are very interesting, which may present future research directions.Reviewer 3 Report
In his review, the authors summarize the current knowledge about on the origin and the plasticity of beige adipocytes in response to different physiological challenges. They particularly focused on the technique of beige adipocyte ablation that they developed as a method to understand the role of postnatal beige adipocytes in adult WAT browning and validate browning candidates in specific genetically modified mice models.
If the authors summarized very well their previous work and the technique they developed about beige adipocyte ablation, the first part is quite vague, lack precisions and some statements need to be revised.
Abstract
l9. The phenomenon of thermogenesis should be defined
l11. What the authors mean by metabolic beneficial?
l16. The authors should be more specific about the “physiological roles” they interrogate with the ablation system
1.Introduction
l21. The authors should revised this statement “When energy intake exceeds energy expenditure, animals store excess energy as fat in adipose and other metabolic tissues.” When energy intake exceeds energy expenditure, the excess of fat is stored in adipose tissue, and this is the main purpose of the tissue. Then, when the storage capacity of the WAT is saturated, the excess of fat is ectopically stored in non adipose organs. This concept named “expandability hypothesis” and has been extensively described.
l24. The authors should defined thermogenesis and what is the exact function of beige adipocytes and UCP1
Beige adipocyte formation: different players
l45. “the actions of sympathetic nerves, the engagement of neurotransmitters and their receptors at the contact sites of the sympathetic nerve and receiving cell, in live mice are most unknown,”. This is not true. The authors should mentioned what is exactly unknown. The role of innervation, tyrosine hydroxylase is the production of NE and the action of NE is well described. Moreover, several papers has been published recently describing the role of “new “neurotransmitters such as NRG4.
Figure1. Figure 1 is currently not informative. The authors should improve the figure and be more specific. Please give more details and examples regarding the physiological stimuli, which immune cells, which hormones, some progenitor markers.
The authors should also add a third arrow mentioning the activation of existing beige adipocytes as they mention it later in the review.
l77. The role of immune cells on thermogenic activity and recruitment of beige adipocytes should be better described. It is not clear how they impact beige functions, positively or negatively?
l81. Once again, the authors should be more specific in which circulating factors they are referring to how they “affect” beige adipocytes.
Beige adipocyte plasticity: Ucp1+ vs Ucp1+-lineage. Is there a mistake in the title?
l104. As mentioned previously, this result suggest that beige adipocytes can be directly activated. This need to be added to Figure 1
l105. “We and other”, the authors need to cite the studies form “other”.
Figure2. The authors should add the gonadal depot
Beige adipocytes in metabolic diseases. The authors should revised this title since they describe thermoneutrality and this status is not a metabolic disease.
l253. The authors should mention the role of Bmp8b in maintaining the activation state of beige adipocytes since transgenic animals overexpressing Bmp8b present browning at 4-5 weeks old mice born and housed at thermoneutrality. And this persisted in adult mice. (Pellegrinelli et al).
Author Response
Abstract
l9. The phenomenon of thermogenesis should be defined
l11. What the authors mean by metabolic beneficial?
l16. The authors should be more specific about the “physiological roles” they interrogate with the ablation system
1.Introduction
l21. The authors should revised this statement “When energy intake exceeds energy expenditure, animals store excess energy as fat in adipose and other metabolic tissues.” When energy intake exceeds energy expenditure, the excess of fat is stored in adipose tissue, and this is the main purpose of the tissue. Then, when the storage capacity of the WAT is saturated, the excess of fat is ectopically stored in non adipose organs. This concept named “expandability hypothesis” and has been extensively described.
l24. The authors should defined thermogenesis and what is the exact function of beige adipocytes and UCP1
We have addressed these concerns in abstract and introduction accordingly.Beige adipocyte formation: different players
l45. “the actions of sympathetic nerves, the engagement of neurotransmitters and their receptors at the contact sites of the sympathetic nerve and receiving cell, in live mice are most unknown,”. This is not true. The authors should mentioned what is exactly unknown. The role of innervation, tyrosine hydroxylase is the production of NE and the action of NE is well described. Moreover, several papers has been published recently describing the role of “new “neurotransmitters such as NRG4.
Similar to neuronal synapse, formation of the sympathetic nerve-receiving cell contact can be very dynamic in respond to cold (hours to days). Release and clearance of neurotransmitters will affect the activation of receiving cells within much short time scale (seconds). Previous and current methodologies focus on the anatomical positioning of sympathetic nerves in the subcutaneous inguinal white adipose tissue (iWAT), for example, by staining the tyrosine hydrolase+ (Th+) sympathetic nerves in fixed tissues. Methods to detect the activities of sympathetic nerves under physiological and pathological conditions heavily rely on bulk detection of norepinephrine, b-adrenergic receptor, or downstream cAMP-PKA signaling in whole tissues, lacking spatial and temporal precision. That is why we are stating that actions of sympathetic nerve in LIVE mice are most unknown.NRG4 has been shown as a neurotrophic factor that can promote axon remodeling, including sympathetic nerves. For example, Pellegrinelli et al (Nature Commun. 2018) showed adipoyte-derived NRG4 can remodel sympathetic nerves to promote WAT browning. But the notion of NRG4 as a neurotransmitter released from sympathetic nerves is not established yet.
Figure1. Figure 1 is currently not informative. The authors should improve the figure and be more specific. Please give more details and examples regarding the physiological stimuli, which immune cells, which hormones, some progenitor markers.
Some details regarding hormones and progenitor marker are added in the figure legend. Our manuscript is specifically focused on the beige adipocyte plasticity, not to comprehensively cover all aspects of the beige adipocyte biology.The purpose to Figure 1 is to setup the stage, describing the major components involved in WAT browning. Too much information regarding immune cells, hormones and progenitor markers in the Figure 1 may diffuse the focus and overwhelm the readers.The authors should also add a third arrow mentioning the activation of existing beige adipocytes as they mention it later in the review.
The arrows are pointing to the different routes of beige adipocyte formation (beige adipogenesis and transdifferentiation), not to different cell types (progenitor, beige or white adipocyte).l77. The role of immune cells on thermogenic activity and recruitment of beige adipocytes should be better described. It is not clear how they impact beige functions, positively or negatively?
This topic has been extensively reviewed in the field. And it is not the key topic of this manuscript.l81. Once again, the authors should be more specific in which circulating factors they are referring to how they “affect” beige adipocytes.
We provide more samples for circulating hormones and metabolites in the figure legend.Beige adipocyte plasticity: Ucp1+ vs Ucp1+-lineage. Is there a mistake in the title?
There is no mistake in the title. "Ucp1+" refers to current state of Ucp1 expression in adipocytes. While "Ucp1+-lineage" refers to the adipocytes that have Ucp1 expression in the past (regardless of the current Ucp1 expression level).l104. As mentioned previously, this result suggest that beige adipocytes can be directly activated. This need to be added to Figure 1
Wolfrum’s study in NCB 2013 demonstrated that Ucp1+ adipocytes can manifest either beige or white adipocyte phenotype, depending on ambient temperature. But this does not prove that beige adipocytes are directly activated by cold through sympathetic nerve innervation.l105. “We and other”, the authors need to cite the studies form “other”.
Reference added.Figure2. The authors should add the gonadal depot
At basal state, there are no substantial amount of Ucp1+ beige adipocytes in the gonadal fat depot. So beige adipocyte ablation will not occur.Beige adipocytes in metabolic diseases. The authors should revised this title since they describe thermoneutrality and this status is not a metabolic disease.
We are not considering thermoneutrality as a metabolic disease state. Here we are describing the necessity to perform metabolic studies under thermoneutrality, in order to determine the contribution of thermogenically activity brown/beige adipocytes to metabolism.l253. The authors should mention the role of Bmp8b in maintaining the activation state of beige adipocytes since transgenic animals overexpressing Bmp8b present browning at 4-5 weeks old mice born and housed at thermoneutrality. And this persisted in adult mice. (Pellegrinelli et al).
We have mentioned BMP8b study as recommended.